# African Swine Fever: Disease Dynamics in Wild Boar Experimentally Infected with ASFV Isolates Belonging to Genotype I and II

**DOI:** 10.3390/v11090852

**Published:** 2019-09-13

**Authors:** Pedro J. Sánchez-Cordón, Alejandro Nunez, Aleksija Neimanis, Emil Wikström-Lassa, María Montoya, Helen Crooke, Dolores Gavier-Widén

**Affiliations:** 1Pathology Department, Animal and Plant Health Agency, APHA-Weybridge, Woodham Lane, New Haw, Addlestone KT15 3NB, UK; alejandro.nunez@apha.gov.uk; 2Department of Pathology and Wildlife Diseases, National Veterinary Institute (SVA), 751 89 Uppsala, Sweden; aleksija.neimane@sva.se (A.N.);; 3Biological Research Center (CIB-CSIC), Ramiro de Maeztu 9, 28040 Madrid, Spain; maria.montoya@cib.csic.es; 4Virology Department, Animal and Plant Health Agency, APHA-Weybridge, Woodham Lane, New Haw, Addlestone KT15 3NB, UK; helen.crooke@apha.gov.uk; 5Department of Biomedical Sciences and Veterinary Public Health, Swedish University of Agricultural Sciences (SLU), Box 7028, 750 07 Uppsala, Sweden

**Keywords:** African swine fever virus, genotype I and II, wild boar, experimental infection

## Abstract

After the re-introduction of African swine fever virus (ASFV) genotype II isolates into Georgia in 2007, the disease spread from Eastern to Western Europe and then jumped first up to Mongolian borders and later into China in August 2018, spreading out of control and reaching different countries of Southeast Asia in 2019. From the initial incursion, along with domestic pigs, wild boar displayed a high susceptibility to ASFV and disease development. The disease established self-sustaining cycles within the wild boar population, a key fact that helped its spread and that pointed to the wild boar population as a substantial reservoir in Europe and probably also in Asia, which may hinder eradication and serve as the source for further geographic expansion. The present review gathers the most relevant information available regarding infection dynamics, disease pathogenesis and immune response that experimental infections with different ASFV isolates belonging to genotype I and II in wild boar and feral pigs have generated. Knowledge gaps in areas such as disease pathogenesis and immune response highlights the importance of focusing future studies on unravelling the early mechanisms of virus-cell interaction and innate and/or adaptive immune responses, knowledge that will contribute to the development of efficacious treatments/vaccines against ASFV.

## 1. General Aspects of African Swine Fever and the Current Situation

Currently, African swine fever (ASF) constitutes the biggest threat faced by the world pork industry in decades. ASF is a devastating haemorrhagic infectious disease which affects domestic and wild suids (all *Sus scrofa*) of all breeds and ages, with high lethality often up to 90–100% in naïve animals. There is no treatment or effective vaccine commercially available. The causative agent, African swine fever virus (ASFV), is a large and complex double-stranded DNA arbovirus that is the only member of the *Asfarviridae* family, genus *Asfivirus* [1]. The molecular phylogeny of the virus is investigated by sequencing the 3’end of the VP72 coding sequence, which differentiates up to 24 distinct genotypes [2].

Among them, the warthog (*Phacochoerus africanus*) has been pointed out as the most important host, while the role of bushpigs (*Potamochoerus larvatus*) in the transmission of ASFV has not been fully elucidated [3]. Soft ticks of the *Ornithodoros* genus, especially *Ornithodoros moubata* and *Ornithodoros erraticus*, have been shown to be both reservoirs and transmission vectors of ASFV [4]. ASFV can survive for long periods in a protein rich environment, remaining infectious for months in refrigerated (4 °C) or frozen blood samples as well as in blood samples kept at room temperature [5]. ASFV remains viable for long periods in faeces and tissues, including uncooked or undercooked pork products. In contrast, the virus is inactivated by heat treatment at 60 °C for 20 min [5,6,7].

ASF is endemic in most sub-Saharan countries in Africa. Transcontinental spread, in which ASFV isolates belonging to genotype I from Western Africa were involved, first occurred to Europe (Spain and Portugal) in 1957 and 1960, and from there to other European countries, South America and the Caribbean. Except for Sardinia, disease was eradicated from outside Africa in the mid-1990s. A second transcontinental spread of genotype II ASFV from Southeast Africa into Georgia occurred in 2007 [8]. ASF subsequently spread to Eastern Europe and later into Western Europe, reaching Belgium in September 2018 [9]. From the initial incursion into Georgia and subsequent spread, the disease affected both domestic pigs and wild boar [10]. African swine fever has led to the deaths of over 800,000 pigs and wild boar across Europe. Initial experimental infections indicated that wild boar are at least as susceptible as domestic pigs initially leading to suggestions that endemicity in wild boar populations might be unlikely [11]. However, the disease established self-sustaining cycles within the wild boar population, a key fact that helped its spread and maintenance in regions. While control efforts may be reducing incidence in domestic pigs, latest reports show a persistent incidence of ASFV-positive wild boar in Belgium and Eastern Europe, representing a significant reservoir that remains a continued threat to the domestic pig industry [12,13]. Increasingly there are reports of antibody-positive wild boar indicating that a proportion of animals survive infection. Some authors have suggested that subclinically infected, chronically infected or survivor pigs might play an important role in disease persistence in endemic areas or in sporadic outbreaks of ASF [14], although the contribution of such animals to virus persistence in a region and their existence is uncertain and under discussion [15]. Moreover, populations of wild boar in many parts of Europe, worryingly continue to expand and increase in abundance, posing a threat via further spread and spill-over to pigs, and increasing the risks of establishing endemic areas of ASF infection [16]. The seriousness of the ASF threat is exemplified by the big jump of ASFV in August 2018, with disease occurring close to the Mongolian borders and later into the world’s largest pig producer, China [17]. Since then, the disease has been out of control, spreading to several provinces and reaching Vietnam, Cambodia, Laos and North Korea in 2019 [18]. While official estimates count 4 million culled hogs, slaughter data suggest 25 times more will be removed from China’s 440 million-strong swine herd in 2019. The United States (U.S) Department of Agriculture forecast in April 2019 a decline of 134 million heads, equivalent to the entire annual output of American pigs. The harm to pigs is especially critical for China, with a USD 128 billion pork industry and the world’s third highest per-capita consumption. China will clearly need to import substantial amounts of pork to satisfy demand, a situation that will impact food prices globally. Wholesale pork prices in China are already 21% higher than a year ago and have risen in the U.S. and European Union after processors sent more of their product to China. It will probably have a lasting effect for several years, moving markets and possibly influencing geopolitical situations [19].

Understanding host–virus interactions and disease dynamics in wild boar by way of experimental infections, not only with genotype II isolates currently circulating in Europe and Asia but also with isolates belonging to other genotypes, is critical for risk assessments and future development of safe and efficacious treatment/vaccines effective in this population. Gathering the most relevant information available and identifying knowledge gaps regarding infection dynamics, disease pathogenesis and immune response, mainly in experimentally infected wild boar, was the aim of the present review.

## 2. Experimental Infections with ASFV Genotype II Isolates in Wild Boar and Disease Progression

Virulence factors, host–virus interactions and the pathogenesis of most of the ASFV genotypes are still far from being understood. This lack of knowledge hampers targeted research into basic mechanisms of disease protection and vaccine design. So far, most studies have been focused on understanding how ASFV isolates of genotype I and II, the only genotypes that have spread outside Africa to date, affect domestic pigs. Since the genotype II outbreak in Georgia in 2007, several studies have been conducted trying to understand infections with particular isolates from Eastern and Central Europe and their disease dynamics. The biological characterization of such isolates has been carried out in domestic pigs [20,21,22,23,24,25,26] and, to a lesser extent, wild boar [11,21,26,27,28] by way of experiments in which different routes of infection (intramuscular, intranasal and contact infections), as well as different low and high infectious doses, were tested (see Table 1). 

In general, wild boar displayed an apparent higher susceptibility to ASFV than domestic pigs in many of the experimental studies. The first oral and intramuscular experimental infections of piglets and adult wild boar with medium (10^3^ HAD_50_ intramuscularly) and high doses (10^6^ TCID_50_ orally) of genotype II isolates from the Caucasus region (Armenia in 2008 and Chechen Republic in 2009) resulted in acute forms of ASF with 100% lethality within less than 10 days [11]. After oral infection, wild boar piglets 9 weeks old (6/6) died between 5 and 7 days, displaying high temperature from day 3–4 post-infection (pi). Apart from haemorrhagic lymph nodes and haemorrhagic gastritis, other macroscopic lesions were not mentioned. Three domestic weaner pigs were placed in contact with the wild boar 2 days after their oral inoculation. Two of the pigs developed acute fatal ASF at 11–12 days after the inoculation of the wild boar, dying 1 week later, while the third pig showed fever at day 20 pi and was euthanized at day 25 pi. One adult wild boar intramuscularly inoculated displayed clinical signs from day 3 pi, dying at day 5 pi, while in-contact adult wild boar (3/3) showed clinical signs at day 8 pi of the intramuscularly inoculated wild boar, dying 2 days later and showing gross lesions characteristic of acute ASF forms (haemorrhages in multiple oedematous and enlarged lymph nodes, hyperplasia of mesenteric lymph nodes, pulmonary hyperaemia and alveolar oedema, haemorrhagic gastritis; no skin lesions described). In a second experiment, one adult wild boar, two adult sows and one piglet orally inoculated (3×10^6^ TCID_50_) with the ASFV Caucasus isolate died or were euthanized between day 8–9 pi. No antibodies were detected in serum samples [27]. Although shedding of ASFV through nasal discharge or faeces seemed to be limited, virus transmission to domestic pigs and wild boar used as in-contact controls was effective, also inducing acute disease in these recipients. Blood was shown to be highly infectious and is likely to be the main source of infection for in-contact pigs [11].

The possibility that low dose genotype II ASFV infections might lead to prolonged incubation times and chronic disease, or the development of a carrier state, was also investigated [21]. Low-dose oronasal infections of domestic pigs and European wild boar were undertaken with the Armenia 2008 isolate. Very low doses of this isolate (3 or 25 haemagglutinating units (HAU), after back titration), which may be equivalent to those obtained by contact with contaminated fomites, swill, carcasses or excretions of infected animals, were sufficient to infect especially weak or runt wild boar by the oronasal route. The time to onset of clinical signs was delayed in these low dose infections, but there were no changes in the course and outcome of infection. Infected pigs developed acute and subacute forms of ASF with severe vascular changes after incubation periods of 11 days to 4 weeks. In wild boar (4–5 months old) inoculated with 100 HAU (25 HAU after back titration), maximum clinical scores were reached from day 13 pi. One animal that displayed clinical signs from day 2 pi died at day 11 pi while the rest displayed clinical signs from day 11 pi, reaching maximum scores from day 13 and dying or being euthanized between day 14 and 17 pi. On the other hand, two wild boar with smaller and weaker appearance inoculated with 25 HAU (3 HAU after back titration) died or were euthanized between day 6 and 10 pi. The rest of animals reached maximum clinical scores from day 14 pi. All animals had to be euthanized approximately 5 to 7 days after onset of first clinical signs (between day 18 and 23 pi). While wild boar showed clinical signs, domestic pigs (8–12 weeks old) inoculated with 25 HAU remained clinically healthy up to day 23 pi, being euthanized within 36 days; clinical signs in domestic pigs inoculated with 3 HAU started from day 12 pi; all animals were euthanized between day 17 and 23 pi. With only a few exceptions, antibody detection yielded negative results. A brief description of macroscopic lesions indicated that haemorrhagic lesions were apparently more severe in wild boar and some individual animals developed secondary infections of the respiratory and gastro-intestinal tract. Thus, results did not suggest the existence of prolonged or chronic individual courses on low-dose infection in wild boar or domestic pigs [21]. 

An ASFV isolate was obtained by animal passage using tissue samples that were DNA positive by PCR. These tissue samples were obtained from a wild boar outbreak with apparently low mortality in North-Eastern Estonia in 2015 [28]. In young European wild boar (4 months old) inoculated by the oronasal route with 10^4.5^ HAU of the North-Eastern Estonia ASFV isolate, nine of ten animals developed clinical signs between day 4 to 6 pi, with worsening clinical signs characteristic of severe acute forms of ASF between day 7–13 pi, when animals were euthanized. These animals also displayed typical ASF lesions of varying severity that increased as the experiment progressed (lung oedema, ebony-coloured gastrohepatic lymph nodes, haemorrhagic and oedematous lymph nodes in all parts of the body, multiple haemorrhages in several organs, severe lung oedema, gall bladder oedema, renal infarction, gastritis and arthritis). A survivor (1 of 10) recovered completely after an acute disease and was commingled with three sentinel wild boar of the same age class from day 50 to 96 post initial inoculation. The sentinels remained healthy and both virus and antibody negative throughout the experiment, so no transmission occurred from the survivor to in-contact pigs. The survivor was negative for ASFV detection in blood and in different tissues taken at day 96 pi, although it had high antibody levels, indicating that virus exposure had occurred [28]. A virus that was recovered from this survivor in the acute phase of the disease was used for additional oronasal inoculations (10^5^ to 10^6.5^ HAU) of twelve adult minipigs, five domestic pigs aged six months and five wild boar, three adults around two years old and two piglets six months old [26]. Three minipigs died between day 8 and 15 pi displaying alveolar oedema, several haemorrhagic lymph nodes and pericarditis, while 9 minipigs (75%) and all domestic pigs (100%) recovered after the acute course of the disease without pathological findings indicative of ASFV infection. However, all adult wild boar succumbed between day 8–9 pi showing severe respiratory distress while both wild boar piglets were euthanized after reaching the humane endpoint between day 16 and 17 pi. Necropsy revealed pulmonary oedema, haemorrhagic lymph nodes and petechiae in the renal cortex. These results suggested that the north-eastern Estonia strain re-isolated from a surviving animal during acute infection showed an attenuated phenotype in minipigs and domestic pigs. However, the higher susceptibility of wild boar is only partially in line with the field observations that showed several apparently healthy, but sero-positive wild boar in the hunting bag of North-Eastern Estonia. Curiously, wild boar piglets survived longer (day 16–17 pi) than adults (day 8–9 pi), which fits with the observation that the detection of antibodies was more likely in the young age class [26]. In a second wild boar experiment with this isolate with three adults and three suckling piglets (unpublished), acute lethal disease was again seen in adults, but the suckling piglets recovered after acute infection [reviewed by 10].

Despite the low mortality rates suggested by some field observations, for example in Estonia [28,29], and the existence of genetic variants of ASFV [30,31], so far the experimental studies mentioned above using different genotype II ASFV isolates from different regions of Eastern and Central Europe, and also now present in Asia, have not demonstrated clear evidence of a reduced virulence in either wild boar or domestic pigs. In most of the cases, a detectable antibody response was not observed due to the rapid onset of acute disease. 

Some studies have suggested the existence of isolates that displayed an attenuated phenotype, especially in domestic pigs [25,26], along with descriptions of occasional survivors [23,28]. However, most of the experiments have demonstrated a limited potential of genotype II ASFV isolates to cause persistent infection and thus generate ASFV carriers. Recently, it has been demonstrated that oral immunization of wild boar with a naturally attenuated non-haemadsorbing ASFV genotype II isolated in Latvia in 2017 (Lv17/WB/Rie1), conferred clinical protection against challenge with homologous high virulence ASFV isolate (Arm07) [32,33]. However, the immunized wild boar were weakly viraemic after challenge with virulent virus and shed it sporadically [33]. Additionally, the Lv17/WB/Rie1 attenuated virus was reported to cause clinical signs, including joint swelling in a previous experiment in domestic pigs [32].

## 3. Other Experimental Infections in Wild Boar and Feral Pigs with ASFV Genotype I Isolates

Early literature also describes preliminary experimental infections in wild boar through inoculation, ingestion and contact using Italian isolates belonging to genotype I (see Table 2). Wild boar displayed high susceptibility and frequently died as a result of such infections, showing characteristic ASF haemorrhagic lesions [34,35]. In a preliminary experiment, one adult boar female and one adult boar male were inoculated subcutaneously in the base of the ear with blood (2 and 4 mL, respectively) containing Tor Sapienza strain (genotype I). An increase in temperature was observed between 24 h (female boar) and 72 h after infection (male boar). Other clinical signs (anorexia, postration and lethargy) were observed in both animals from day 5 pi. Female boar also displayed haemorrhagic diarrhoea from day 8 pi and died at day 11 pi while male boar died at day 13 pi. A detailed macroscopic evaluation described the presence of typical ASF haemorrhagic lesions equivalent to those observed in domestic pigs such as occasional subcutaneous petechial haemorrhages, intraparietal diffuse petechial haemorrhages in the abdominal wall, mild hydropericardium with sero-haemorrhagic fluid, petechial haemorrhages in endocardium, diffuse pulmonary congestion along with alveolar oedema and presence of subpleural and parenchymal petechiae, mild ascites, hyperaemic splenomegaly, mild hepatic congestion and hepatomegaly, oedema in the gallbladder wall with presence of petechial haemorrhages on the mucosal surface, reddish fibrin deposits on the serosa surface of the cecum and colon and haemorrhages on the mucosal surface of the rectum. The kidneys showed petechiae in renal cortex, with the medulla and the renal pelvis appearing uniformly haemorrhagic. The female boar also displayed a sero-haemorrhagic fluid with fibrin deposits within the breast cavities. Regarding the lymph nodes, the mesenteric and bronchial were enlarged and strongly hyperaemic while the gastrohepatic and inguinal were severely enlarged and seemed like blood clots, displaying a strong dark red colour [34].

In a second experiment, four young wild boar (two males and two females) were split in two pens, each containing two boar of different sex. In both groups, the male boar received 2 mL of leukocytes culture infected with Nemi strain (genotype I) by intramuscular (group 1) and oral route (group 2) while female boar were kept in close contact with the infected animals. In group 1, the male boar intramuscularly inoculated displayed an increase in temperature from 24 h pi and died at day 11 pi showing less severe typical ASF macroscopic lesions (focal subcutaneous haematomas, mild hydropericardium, occasional petechial haemorrhages in renal cortex, alveolar oedema and systemic lymphadenitis) than observed in animals infected intramuscularly in the first experiment. Regarding the female boar in contact, clinical signs, evidence of viral infection or macroscopic lesions characteristic of ASF were not observed throughout several weeks after the death of the male boar. In group 2, the male boar infected orally developed clinical signs from day 6 pi, being euthanized in a moribund state at day 20 pi, while the female boar in contact developed clinical signs from day 20 pi and was euthanized in a moribund state at day 28 pi. In both animals, macroscopic lesions characteristic of ASF were scarce and mild showing mild alveolar oedema and systemic lymphadenitis [35]. 

In more recent experimental studies, four wild boar were inoculated intramuscularly with 10^6^ HAU of Sardinian 2008 ASFV isolate (genotype I). Animals displayed respiratory distress and were euthanized between day 5 and 8 pi after reaching the humane endpoint. The peak temperature was detected at day 4 pi (41.7 °C), while the highest clinical scores were observed at day 8 pi. At necropsy pulmonary oedema, haemorrhagic lymph nodes, hepatic congestion, gall bladder wall oedema and haemorrhages were the most common macroscopic lesions [36].

An interesting study, in which 7-week-old classical swine fever virus postnatal persistently infected (CSFV PI) wild boar (*n* = 3) and wild boar tested pestivirus-free (*n* = 3) were experimentally infected by intramuscular route with a dose of 10^4^ TCID_50_ of ASFV E75 isolate (genotype I), revealed that only CSFV PI wild boar showed a progressive clinical disease from day 4 pi with external lesions and clinical signs typically described in acute forms of ASF (high temperature, dyspnoea, tremors, cyanosis and petechial haemorrhages in the ears and abdomen) [37]. While CSFV RNA load remained unaltered over the study, the levels of type I IFNα and IL-10 in sera were almost undetectable and CD4+ T-cells displayed a significant decrease. Survival rates were similar in both groups. CSFV PI wild boar (3/3) died between day 6 and 7 pi, while pestivirus-free ASFV-infected wild boar died suddenly at day 8 pi (2/3), and the third animal was euthanized at day 10 pi, with all pestivirus-free animals showing only mild clinical signs of ASF. Apart from the wild boar euthanized at day 10, ASFV-specific antibodies were not detected in any of the animals. In both groups, wild boar displayed characteristic macroscopic lesions of acute ASF during necropsy. The survival rate following ASFV infection was similar in both experimental groups as was ASFV DNA load, which suggested that ASFV infection does not produce any interference with persistent CSFV replication, or vice versa. However, the immunosuppression state in CSFV PI wild boar may have influenced ASF progression and development of haemorrhagic clinical signs [37]. 

Biological characterization of ASFV genotype I strains from the Iberian Peninsula (highly virulent isolate Lisbon 1960) and the Dominican Republic (moderate virulent isolate DR 1979) were also carried out through intranasal inoculations of feral pigs (*Sus scrofa*) trapped in Florida, USA [38]. Twelve feral pigs were split in two experimental groups. Two members of each group were inoculated by instilling into each nostril and into the mouth a total dose of approximately 10^7^ HAU_50_ of Lisbon 1960 or DR 1979 respectively, while the rest of animals (four per group) were kept in close contact. Clinical signs (temperatures up to 41.1 °C, lethargy, inappetence) were observed in both groups of inoculated pigs between day 3 and 4 pi. All pigs infected with Lisbon 1960 isolate were either dead or moribund 2 to 6 days after the onset of fever (day 7–8 pi for inoculated pigs and day 14–20 pi for contact pigs), developing an acute form of ASF. All pigs exposed to DR 1979 isolate also died but survived 6 to 13 days after the onset of fever (day 11–16 pi for inoculated pigs and day 19–22 pi for contact pigs), developing a subacute form of ASF. In a previous experiment, only 20% of domestic pigs of a similar age died after infection with DR 1979 isolate [39]. Although pigs infected with Lisbon 1960 (inoculated and in contact) developed an acute clinical form of ASF, the severity of macroscopic lesions described corresponded to subacute forms of ASF characterized by the presence of extensive and severe haemorrhagic lesions [38].

These included the presence of extensive areas of purplish skin discoloration in ventral areas, severe hydropericardium with red-tinged fluid, hyperaemic splenomegaly, petechiae in renal cortex and extensive haemorrhages in renal pelvis and heart, generalized haemorrhagic lymphadenitis (especially in mandibular, gastrohepatic, renal and inguinal lymph nodes), severe ascites with red-tinged fluid, perirenal oedema and marked retroperitoneal haemorrhages. The haemadsorption reaction, and the direct and indirect immunofluorescence were used to confirm the presence of ASFV in the blood and selected tissues samples from all infected pigs. ASFV-specific antibody response was only detected in pigs that died from 5 to 13 days after the onset of clinical signs [38].

## 4. Conclusions and Future Research

While control efforts, particularly increased biosecurity and professionalisation of the pig industry, have successfully reduced the incidence of ASF in domestic pigs, the wild boar population represents a substantial reservoir in Europe and probably also in Asia, which will hinder eradication and serve as a source for further geographic expansion [40]. Therefore, understanding host–virus interactions and disease dynamics in wild boar experimentally infected with genotype II isolates currently circulating in Europe and Asia is critical for risk assessments and vaccine development. To date, relatively few ASFV experimental infections have been carried out in wild boar and therefore the body of data is not comparable to the one in domestic pigs. However, results in both domestic pigs and wild boar have highlighted variability relating to experimental outcomes and severity of signs after experimental infections with the currently circulating ASFV genotype II isolates in Europe, possibly above the expected biological variability, and attributable to the different experimental settings. In any case, except for the naturally attenuated ASFV isolate found in Latvia in 2017 (Lv17/WB/Rie1), evidence for reduced virulence in the rest of the ASFV isolates assessed was not observed. In most of the experimental infections with genotype II isolates, wild boar of different ages displayed a high susceptibility to ASFV and disease development. In several studies, the susceptibility was even higher than that shown by domestic pigs of the same ages. Such differences in the course and severity of ASFV infection and development of disease between domestic pigs and Eurasian wild boar may suggest subtle differences in the pathogenetic mechanisms between the two subspecies that future studies should elucidate.

Due to the close taxonomic relationship between wild boar and domestic pigs it could be assumed that ASFV infection in wild boar has a similar course as in domestic pigs, and furthermore, that an ASF vaccine developed for domestic pigs can be applied to vaccinate wild boar with similar results. However, even though both domestic pigs and European wild boar are of the same species (*Sus scrofa*), they belong to different subspecies. Besides genetic differences, domestic pigs are managed and therefore their diet, reproduction and health status are controlled, whereas free-ranging wild boar are subjected to natural nutritional, climatic and reproductive variations and cycles. Wild boar in nature show intraspecific aggression, carry a mixed pathogen load and suffer periods of stress, which may compromise the function of their immune system. All together, these factors may affect the outcome of ASFV infection in wild boar. Therefore, it is important to study the pathogenetic and immunological mechanisms of ASF specifically in wild boar so that not only biological differences with domestic pigs can be identified but also additional differences between experimental and natural ASFV infection in wild boar can be assessed. These data can contribute to epidemiological studies, modelling and design of control strategies such as vaccine development and deployment. The fact that information about the presentation of ASF in wild boar was scarce, and derived only from limited experimental studies, led to early assumptions that it was unlikely that Caucasian isolates would become endemic in European wild boar populations without a significant change in virulence [11]. The evolution of ASF in Europe shows otherwise and emphasises the need for more studies.

In most of the studies included in the present review, clinical evaluations were carried out by way of equivalent clinical scoring systems, which allowed to set up accurately the evolution and duration of clinical course of disease. In addition, viraemia levels and spread mechanisms through different organic fluids and routes of excretion were determined for swabs and blood samples by equivalent molecular detection techniques. Also, virus burdens in target tissues taken once the animals reached the humane endpoint were precisely evaluated. However, the published studies are more limited on precise, systematic and semi-quantitative evaluations of macroscopic and/or histopathologic lesions, which may establish differences among individual animals. In addition, there are few comparative evaluations between pig and wild boar describing the characteristics of the disease induced by experimental infection with ASFV.

Furthermore, apart from ASFV-specific antibody responses, other innate or adaptive immunological parameters that may contribute to memory responses, key for vaccine design, have not been extensively assessed in wild boar. As in domestic pigs, no correlates of protection in wild boar are defined for ASFV infection, which can only be informed by a deeper knowledge of the host–virus interaction. Thus, experimental studies of ASFV in domestic pigs and wild boar require approaches that should be assessed not only by virologists, molecular biologists and immunologists, but also by pathologists. Therefore, along with standard protocols to evaluate virological parameters and immunological responses, scientists involved in ASFV studies would benefit from applying standard pathological evaluation protocols for adequate individual, intergroup and inter-experiment comparison. In this sense, standardized scoring systems to perform macroscopic and histopathological assessments of ASFV-inoculated pigs have been proposed [41]. Such evaluation protocols have been demonstrated to be a useful tool for the identification and evaluation of lesions severity, data collections and analysis in the course of ASFV experiments focused on testing new vaccine candidates [42,43,44,45] or during pathogenesis studies and/or the biological characterization of new ASFV isolates [46]. 

Although pathogenetic mechanisms induced by high and moderate virulence ASFV isolates of genotype I have been widely studied since the disease first appeared in the Iberian Peninsula in the 1950–60s [47,48], less is known about such mechanisms or the immune response induced by isolates of genotype II in domestic pigs and, to a lesser extent, wild boar despite their high susceptibility to disease. Comparative time-course studies would provide novel and more detailed information which is currently lacking on the pathogenesis and immunology of ASF in domestic pigs and wild boar. Early studies have shown that in natural infections and experimental infections by intranasal route, ASFV enters preferably via the tonsils or dorsal pharyngeal mucosa and then extends to mandibular or retropharyngeal lymph nodes. After extensive replication in lymphoid tissues, ASFV spreads throughout the body disseminated by leukocytes and/or erythrocytes via lymphatic fluid or blood [49,50,51]. Future studies should be focused on unravelling, from the very early stages after infection, pathogenic mechanisms of ASFV genotype II isolates, virus–cell interactions and local immune responses in organs positioned at the opening of respiratory and gastrointestinal tract (palatine tonsils, lung and regional lymph nodes). This will contribute to our understanding of infection dynamics and disease pathogenesis and will help towards the development of safe and efficacious treatments/vaccines against ASFV.

## Figures and Tables

**Table 1 viruses-11-00852-t001:** Experimental infections of wild boar with African swine fever virus (ASFV) Genotype II isolates. Information about experimental conditions of domestic pigs that were infected in parallel with wild boar have also been provided. (GT): genotype; (WB): wild boar; (DP): domestic pigs; (HAU): haemagglutinating units; (TCID): Tissue Culture Infectious Dose (TCID), (ND): Not determined; (NA): Not administered (in contact animals).

Isolate/ Origin	GT	Type /Number of Animals	Estimated Age	Dose	Route of Exposure	Onset of Clinical Signs After Infection	Survival After Infection (dpi)	Ref.
Armenia 2008	II	WB (*n* = 6)	9 weeks	10^6^ TCID	Oral	3–4 dpi (6)	5–7 dpi (6)	[11]
	II	DP (*n* = 3)	Weaner pigs	NA	In contact	11–12 dpi (2)/ 20 dpi (1)	17 dpi (2)/ 25 dpi (1)	
Chechen Republic 2009	II	WB (*n* = 1)	9 months	10^3^ HAU	Intramusc.	3 dpi (1)	5 dpi (1)	[11]
	II	WB (*n* = 3)	9 months	NA	In contact	8 dpi (3)	10 dpi (3)	
Caucasus isolate	II	WB (*n* = 1)	10 years	3 × 10^6^ TCID	Oral	ND	8–9 dpi (4)	[27]
		Sow (*n* = 2)	4–5 years					
		WB (*n* = 1)	Piglet					
Armenia 2008	II	WB (*n* = 6)	4–5 months	100 HAU (25 HAU after back titration)	Oronasal	WB: 2–5 dpi (1)/ 11–13 dpi (5)	WB: 11 dpi (1)/ 14–17 dpi (5)	[21]
		DP (*n* = 6)	8–12 weeks			DP: 23 dpi (1)/ 30–33 dpi (5)	DP: 28 dpi (1)/ 34–36 dpi (5)	
	II	WB (*n* = 6)	4–5 months	10 HAU (3 HAU after back titration)	Oronasal	WB: 0–9 dpi (2; runt animals)/ 14–19 dpi (4)	WB: 6–10 (2; runt animals)/ 18–23 dpi (4)	
		DP (*n* = 6)	8–12 weeks			DP: 12–19 dpi (6)	DP: 17–23 dpi (6)	
North-Eastern Estonia	II	WB (*n* = 10)	4 months	10^4.5^ HAU	Oronasal	4–6 dpi (10)	7–13 dpi (9)/ recovered WB (1)	[28]
North-Eastern Estonia	II	1 recovered WB and 3 WB (sentinels)	5 months	NA	In contact	No clinical signs (4)	End of trial at 96 dpi. All animals (4) completely healthy.	[28]
North-Eastern Estonia	II	Minipigs (*n* = 12)	6 months	10^5^ HAU	Oronasal	Minipigs: 7 dpi (12)	Minipigs: 8–15 dpi (3)/recovered minipigs (9)	
		DP (*n* = 5)	6 months			DP: 4–6 dpi (4)/10 dpi (1). All animals without clinical signs from 19 dpi	DP: All animals recovered (5)	[26]
	II	WB (*n* = 3)	2 years (adults)	10^6.5^ HAU	Oronasal	Adults WB: 3–4 dpi (3)	Adult WB 8–9 dpi (3)	
		WB (*n* = 2)	6 months (piglets)			Piglets WB: 3–4 dpi (2)	Piglets: 16–17 dpi (2)	

**Table 2 viruses-11-00852-t002:** Experimental infections of wild boar and feral pigs with ASFV Genotype I isolates. (GT): genotype; (WB): wild boar; (DP): domestic pigs; (HAU): haemagglutinating units; (TCID): Tissue Culture Infectious Dose (TCID), (ND): Not determined; (NA): Not administered (in contact animals).

Isolate/ Origin	GT	Type /Number of Animals	Estimated Age	Dose	Route of Exposure	Onset of Clinical Signs After Infection	Survival After Infection (dpi)	Ref.
Tor Sapienza	I	WB (*n* = 2)	Adults	ND (2–4 mL infected blood	Subcutaneous (base of the ear)	Temperature: from 24–72 h/clinical signs: from 5 dpi (2)	11–13 dpi (2)	[34]
Nemi	I	WB (*n* = 1)	Young	ND (2 mL of leukocyte culture infected)	Intramusc. (neck)	Temperature: from 24 h (1)	11 dpi (1)	[35]
		WB (*n* = 1)			In contact	No clinical signs (1)	Euthanized weeks after infection	
	I	WB (*n* = 1)	Young	ND (2 mL of leukocyte culture infected)	Oral	6 dpi (1)	20 dpi (1)	[35]
		WB (*n* = 1)			In contact	20 dpi (1)	28 dpi (1)	
Sardinian 2008	I	WB (*n* = 4)	ND	10^6^ HAU	Intramusc.	3–4 dpi (4)	5–8 dpi (4)	[36]
E75	I	CSFV PI WB (*n* = 3)	7 weeks	10^4^ TCID	Intramusc.	CSFV PI WB: 4 dpi (3)	CSFV PI WB: 6–7 dpi (3)	[37]
		Pestivirus-free WB (*n* = 3)				Pestivirus-free WB: 4 dpi (3)	Pestivirus-free WB: 8–10 dpi (3)	
Lisbon 1960	I	Feral pigs (2)	Adults	10^7^ HAU	Intranasal	Feral pigs: 3–4 dpi (2)	Feral pigs: 7–8 dpi (2)	[38]
	I	Feral pigs (4)	Adults	NA	In contact	Feral pigs: 8–17 dpi (4)	Feral pigs: 14–20 dpi (4)	
Dominic. Republic 1979	I	Feral pigs (2)	Adults	10^7^ HAU	Intranasal	Feral pigs: 3–4 dpi (2)	Feral pigs: 11–16 dpi (2)	[38]
	I	Feral pigs (4)	Adults	NA	In contact	Feral pigs: 10–13 dpi (4)	Feral pigs: 19–22 dpi (4)	

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
