# Peer review of "African Swine Fever: Disease Dynamics in Wild Boar Experimentally Infected with ASFV Isolates Belonging to Genotype I and II"

_viruses, 2019, doi:10.3390/v11090852_

Round 1
Reviewer 1 Report
The short review by Sánchez-Cordón et al., addresses issues related to the current outbreaks of African swine fever in Europe and Asia (caused by a genotype II virus) and the earlier incursion of the genotype I virus initially into the Iberian peninsula (which is still present in Sardinia). The review covers the role of wild boar and gathers information on results from experimental studies as well. Some of the information presented is not supported by citations (why not?) and it would be helpful for readers to be able to refer to the primary source of the information.
There are some minor changes that could be made to correct errors and improve clarity:
Line 42. It is mixing terminology to refer to “the C-terminal end of the VP72 gene”. It is proteins that have a C-terminus. The sequencing is of the 3´ end of the VP72 coding sequence. Line 46, “warthogs and bushpigs are pointed as the most important hosts”, not supported by citation and e.g. Penrith et al., 2019 DOI: 10.1111/tbed,13117 mention warthogs as the most important host and the role of bushpigs in the transmission of ASFV not to be fully elucidated The text on lines 76-79 reads: “While official estimates count 4 million culled hogs, slaughter data suggest 25 times more will be removed from China's 440 million-strong swine herd in 2019. The United States (U.S) Department of Agriculture forecast in April 2019 a decline of 134 million heads, equivalent to the entire annual output of American pigs.” This information is useful and relevant but is currently unreferenced. The source of the information should be given. Line 90, “identify” should be “identifying” Line93 “Experimental infections of ASFV genotype II”….. should be “Experimental infections with ASFV genotype II” Line 150, I know what “virus PCR positive tissue samples” means but I think it is unfortunate terminology. Presumably, the tissue samples were determined to be ASFV DNA positive by PCR. I hope the terminology for virus titres (as HAU) can be corrected, throughout the text the superscripts are absent currently, e.g. on line 165, 105 HAU really means 10 E5 or 105 HAU. The Tables need slight reformatting so that words are not split. In Table 1, for the North Eastern Estonia isolate, in the right hand column, the text should be DP: All animals recovered (delete “were”) The authors use the term “wild boar” and “wild boars” interchangeably, I think wild boar should be used Line 241, it should be “inoculated pigs” not “innoculated pig” Line 244, what is the reference for the “previous experiment with DR’79”? The following text is unclear, it seems to suggest infection by the DR’79 isolate in pigs inoculated with the Lisbon’60 isolate. Is this correct? Line 280, it should be “domestic pigs” not just “domestic”; and “it is understandable to assumed” could be changed to “it could be assumed” Line 281, 282 and 283 it should be “domestic pigs” not just “domestic” Line 308, it should be “antibody responses” Line 336, “what” should be “which”
Author Response
Reviewers' comments:
We would like to thank the Reviewers as well as the Editor for their comments and suggestions which have enabled to improve substantially the original manuscript. The answer to reviewer’s comments can be find below. Along with these changes, we have also completed the original manuscript by adding extra data from preliminary experimental infections in wild boar that used Italian isolates belonging to genotype I. These data have been also included in table 2.
Review 1
The short review by Sánchez-Cordón et al., addresses issues related to the current outbreaks of African swine fever in Europe and Asia (caused by a genotype II virus) and the earlier incursion of the genotype I virus initially into the Iberian peninsula (which is still present in Sardinia). The review covers the role of wild boar and gathers information on results from experimental studies as well. Some of the information presented is not supported by citations (why not?) and it would be helpful for readers to be able to refer to the primary source of the information.
There are some minor changes that could be made to correct errors and improve clarity:
Line 42. It is mixing terminology to refer to “the C-terminal end of the VP72 gene”. It is proteins that have a C-terminus. The sequencing is of the 3´ end of the VP72 coding sequence.
Following the reviewer’s suggestion, the sentence has been changed as follows: “The molecular phylogeny of the virus is investigated by sequencing the 3´end of the VP72 coding sequence, which differentiates up to 24 distinct genotypes”
Line 46, “warthogs and bushpigs are pointed as the most important hosts”, not supported by citation and e.g. Penrith et al., 2019 DOI: 10.1111/tbed,13117 mention warthogs as the most important host and the role of bushpigs in the transmission of ASFV not to be fully elucidated
Following the reviewer’s suggestions, the sentence has been amended as follows: “Among them, the warthog (Phacochoerus africanus) has been pointed as the most important host while the role of bushpigs (Potamochoerus larvatus) in the transmission of ASFV has not been fully elucidated (Penrith et al., 2019). This new citation has been added to the text and the references list: [3] Penrith, M.L., Bastos, A.D., Etter, E.M.C., Beltrán-Alcrudo, D. Epidemiology of African swine fever in Africa today: Sylvatic cycle versus socio-economic imperatives. Transbound Emerg Dis. 2019, 66, 672-686.
The text on lines 76-79 reads: “While official estimates count 4 million culled hogs, slaughter data suggest 25 times more will be removed from China's 440 million-strong swine herd in 2019. The United States (U.S) Department of Agriculture forecast in April 2019 a decline of 134 million heads, equivalent to the entire annual output of American pigs.” This information is useful and relevant but is currently unreferenced. The source of the information should be given.
The source of the information was provided in the text (originally as reference 18) and the references list: Dan Murtaugh, D., Curran, E. Pig ‘Ebola’ virus sends shock waves through global food chain. 3 May 2019. https://www.bloomberg.com/news/articles/2019-05-02/pig-ebola-virus-sends-shock-waves-through-global-food-chain
Line 90, “identify” should be “identifying”.
It has been amended
Line93 “Experimental infections of ASFV genotype II”….. should be “Experimental infections with ASFV genotype II”
It has been amended
Line 150, I know what “virus PCR positive tissue samples” means but I think it is unfortunate terminology. Presumably, the tissue samples were determined to be ASFV DNA positive by PCR.
Following the reviewer’s suggestions, the sentence has been re-written as follows: “An ASFV isolate was obtained by animal passage using tissue samples that were DNA positive by PCR. These tissue samples were obtained from a wild boar outbreak with apparently low mortality in North-Eastern Estonia in 2015”
I hope the terminology for virus titres (as HAU) can be corrected, throughout the text the superscripts are absent currently, e.g. on line 165, 105 HAU really means 10 E5 or 105 HAU. The Tables need slight reformatting so that words are not split.
In the original version of the manuscript that we uploaded, all superscripts throughout the text were fine. Regarding the tables, the same issue. The tables uploaded did not show any word split. This problem may be corrected by adjusting the width of columns. We have realized that it is a common mistake only present in the version sent to the reviewers. We hope these mistakes be corrected properly during editing process. For now this is a problem beyond our control.
In Table 1, for the North Eastern Estonia isolate, in the right hand column, the text should be DP: All animals recovered (delete “were”)
It has been amended.
The authors use the term “wild boar” and “wild boars” interchangeably, I think wild boar should be used
The term “wild boar” is now used throughout the manuscript both for the singular and plural form.
Line 241, it should be “inoculated pigs” not “innoculated pig”
It has been amended.
Line 244, what is the reference for the “previous experiment with DR’79”?
The reference (Mebus et al., 1979) has been added in the manuscript and the references list: [39] Mebus, C.A., Dardiri, A.H. Additional characteristics of disease caused by the African swine fever viruses isolated from Brazil and the Dominican Republic. Proceedings of the 83rd Annual Meeting of the United States Animal Health Association, San Diego, California, United States of America, 28th October-2nd November, 1979, 227-239.
The following text is unclear, it seems to suggest infection by the DR’79 isolate in pigs inoculated with the Lisbon’60 isolate. Is this correct?
We agree with the reviewer. This paragraph was incorrectly written. The paragraph has been re-written as follows: “Although pigs infected with Lisbon’60 (inoculated and in contact) developed an acute clinical form of ASF, the severity of macroscopic lesions described corresponded to subacute forms of ASF characterized by the presence of extensive and severe haemorrhagic lesions”.
Line 280, it should be “domestic pigs” not just “domestic”; and “it is understandable to assumed” could be changed to “it could be assumed”
The changes suggested have been carried out: “Due to the close taxonomic relationship between wild boar and domestic pigs it could be assumed that ASFV infection in wild boar has a similar course as in pigs”.
Line 281, 282 and 283 it should be “domestic pigs” not just “domestic”
We agree with the reviewer. The change suggested has been made.
Line 308, it should be “antibody responses”
It has been amended.
Line 336, “what” should be “which”
It has been amended.
Reviewer 2 Report
The manuscript is a review of African Swine Fever outbreaks in Europe following spread from Africa.
The paper is extremely well written, follows a logical path of exposition and delivers highly pertinent and valuable information on the background of this disease and its spread across Europe and into China.
In my opinion it can be accepted as it is although there are a minimal number of typographical corrections that would be advisable. I have noted these below:
Table 1 – width of first column needs adjustment
L196 text justification has spread this line out strangely
L261 needs “the” added just before “pig industry”
L266 “vaccines” should be “vaccine”
L280 “assumed” should be “assume”
L294 “limit” should be “limited”
L336 suggest split this sentence into two e.g. “…lymph nodes). This will contribute….”
Overall an excellent and informative article.
Author Response
Reviewers' comments:
We would like to thank the Reviewers as well as the Editor for their comments and suggestions which have enabled to improve substantially the original manuscript. The answer to reviewer’s comments can be find below. Along with these changes, we have also completed the original manuscript by adding extra data from preliminary experimental infections in wild boar that used Italian isolates belonging to genotype I. These data have been also included in table 2.
Review 2
The manuscript is a review of African Swine Fever outbreaks in Europe following spread from Africa.
The paper is extremely well written, follows a logical path of exposition and delivers highly pertinent and valuable information on the background of this disease and its spread across Europe and into China.
In my opinion it can be accepted as it is although there are a minimal number of typographical corrections that would be advisable. I have noted these below:
Table 1 – width of first column needs adjustment
This is an issue that reviewer 1 has also indicated. In the original version of the manuscript that we uploaded, the width of columns was fine. The tables uploaded did not show any word split. We have realized that it is a mistake only present in the version sent to the reviewers. We hope the width of columns be corrected properly during editing process. For now this is a problem beyond our control.
L196 text justification has spread this line out strangely
As we have mentioned above, these are issues that should be sorted out during the editing process. We will mentions such issues to the editor in order to correct any mistake.
L261 needs “the” added just before “pig industry”
It has been amended
L266 “vaccines” should be “vaccine”
It has been amended
L280 “assumed” should be “assume”
The sentence has been changed as follows: “Due to the close taxonomic relationship between wild boar and domestic pigs it could be assumed that ASFV infection in wild boar has a similar course as in domestic pigs
L294 “limit” should be “limited”
It has been amended
L336 suggest split this sentence into two e.g. “…lymph nodes). This will contribute….”
Following the reviewer’s suggestion, the sentence has been split into two.
Overall an excellent and informative article.